# DACO: Towards Application-Driven and Comprehensive Data Analysis via Code Generation

**Xueqing Wu[1], Rui Zheng[2], Jingzhen Sha[1], Te-Lin Wu[1], Hanyu Zhou[1], Mohan Tang[1],**
**Kai-Wei Chang[1], Nanyun Peng[1], Haoran Huang[3]**
[1]University of California, Los Angeles      [2] Fudan University      [3] ByteDance

## Abstract

Data analysis is a crucial analytical process essential for deriving insights from real-world databases. As shown in Figure 1, the need for data analysis typically arises from specific application scenarios, and requires diverse reasoning skills including mathematical reasoning, logical reasoning, and strategic reasoning. Existing work often focus on simple factual retrieval or arithmetic resolutions and thus are insufficient for addressing complex real-world queries. This work aims to propose new resources and benchmarks on this crucial yet challenging and under-explored task. Due to the prohibitively high cost of collecting expert annotations, we use large language models (LLMs) enhanced by code generation to automatically generate high-quality data analysis, which will later be refined by human annotators. We construct the **DACO dataset**, containing (1) 440 databases (of tabular data) collected from real-world scenarios, (2) $\sim 2k$ automatically generated query-answer pairs that can serve as weak supervision for model training, and (3) a concentrated but high-quality test set with human refined annotations that serves as our main evaluation benchmark. Experiments show that while LLMs like GPT-4 exhibit promising data analysis capabilities, they are still evaluated as less helpful than human-written analysis on 58.1% cases. Leveraging our weak supervision data, we experiment with various fine-tuning methods, including supervised fine-tuning (SFT) and reinforcement learning from human feedback (RLHF). Our trained model outperforms existing baselines for table question answering, and RLHF further boosts the helpfulness of generated analysis on 58.5% cases. Data and code are released at `https://github.com/shirley-wu/daco`.

## 1 Introduction

Data analysis is the process of systematically applying statistical and logical reasoning to comprehend data and derive insights. Existing literature on table question answering has investigated answering queries about information given by structural data (*e.g.*, tables) [7, 26, 24]. However, they either focus straightforward factual retrieval or short-form arithmetic resolutions over retrieved entities while real-world data analysis can involve more complex analytical processes.

Take the scenario in Figure 1 as an example: a user is investigating potential age discrimination of a shop. To effectively answer queries such as this one, a chain of mathematical and logical reasoning and interacting with the data is required. For instance, *finding 1* is inferred from the membership data ('member' table) through mathematical and analytical reasoning, while *finding and suggestion 2* are derived from both 'member' and 'happy_hour_member' tables through mathematical and strategic reasoning. These rigorous quantitative analyses eventually conclude the opposite to the user's hypothesis. As valuable as the conclusive suggestions such comprehensive analysis can bring, the extensive labor-efforts, hinted by these examples, can hinder the efficiency of gaining intelligence

38th Conference on Neural Information Processing Systems (NeurIPS 2024) Track on Datasets and Benchmarks.

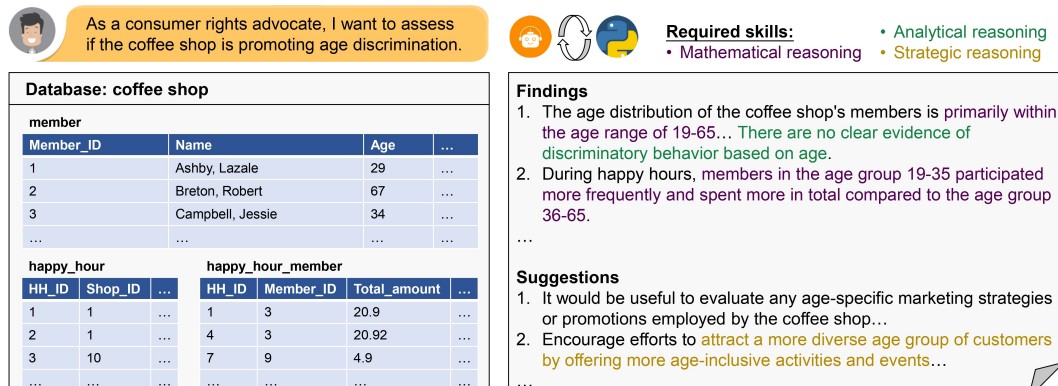

Figure 1: **Task overview.** Given a user query driven by an application scenario, a data analysis system should produce an answer containing findings and suggestions based on the database. This requires the system to perform mathematical, logical and domain-specific reasoning, which can be done through invoking external tools such as Python libraries. In this example, *finding 1* is inferred from analyzing age distribution within the membership data ('member' table) through mathematical reasoning and analytical reasoning. *Finding 2* is inferred by comparing the ages of the happy hours participants (using 'member' and 'happy_hour_member' tables) through mathematical reasoning, and *suggestion 2* is further derived by relating the data to coffee shop business setting through strategic reasoning.

from the data in a competitive business environment. It is thus imperative to devise a system that is able to automate the aforementioned data analysis process.

To this end, we introduce a new dataset for this challenging task, DACO, **d**ata **a**nalysis via **co**de generation. DACO is constructed from a set of diverse real-world databases associated with curated user queries. In light of the previously described labor-intensive challenge, we propose to leverage LLMs with a multi-turn chained prompts to automatically curate the analytical answers for each query. Specifically, our designed framework employs the code generation capabilities of GPT-4 [28] for automating the statistical analysis, interleaved with its ability to interpret the obtained quantitative results. The DACO dataset contains $440$ databases and $1,942$ associated user queries, where each query is annotated with an average of $3.3$ coding steps and $1.9k$ lines of code during intermediate steps and a final output of $\sim 10$ bullet points. This resource can be used for both model fine-tuning and evaluation. To provide a refined benchmarking resource, we curate a high-quality test set through comprehensive human annotations on a subset of 100 samples. Detailed statistics are in Table 1.

We evaluate three types of methods on this new dataset: (1) existing methods designed for table QA tasks, (2) prompt-based LLMs, and (3) fine-tuned LLMs. We use **helpfulness** as our main evaluation metric, assessed through pair-wise comparison. We observe that proprietary LLMs such as GPT-4 demonstrate strong data analysis capabilities and significantly outperform table QA models. However, they still fall short of human performance by $58.1\%$ in pair-wise evaluations. Regarding fine-tuned LLMs, via supervised fine-tuning (SFT) using automatically generated annotations that include both code generation trajectories and the final answers. Inspired by the recent success of reinforcement learning from human feedback (RLHF) [29, 35, 4, 5, 48, 38], we employ RLHF to further align the SFT models with human preferences towards *helpful* data analysis. Our SFT model exhibits promising data analyais capabilities and outperforms table QA baselines despite lagging behind proprietary LLMs. On top of SFT, RLHF further enhances the output helpfulness in $58.5\%$ of cases.

In summary, our contributions are as follows: (1) We explore the challenging task of data analysis, where we construct the DACO dataset with our proposed multi-turn prompting technique on a diverse set of real-world databases. (2) We curate a human-refined evaluation set for benchmarking models. (3) We evaluate a diverse set of models on this challenging dataset, including existing models designed for table QA, prompt-based LLMs, and fine-tuned LLMs with both SFT and RLHF. (4) Our dataset and code are made publicly available at `https://github.com/shirley-wu/daco`.

## 2 Task Formulation

As shown in Figure 1, the input to our task consists of a database $\mathcal{D}$ and a query $\mathbf{q}$, where the database

$\mathcal{D}$ is a relational database containing multiple named tables. The output $\mathbf{y}$ is formatted as two lists: findings and suggestions.

Inspired by recent work on tool usage and LLM agent [40, 41, 23], we allow the LLM to invoke tools for multiple turns before producing the final output $\mathbf{y}$. At the $i$-th turn, the *action* $\mathbf{a}_i$ is generated by the LLM, and the *observation* $\mathbf{o}_i$ is produced by the environment after executing the corresponding action $\mathbf{a}_i$. While the action $\mathbf{a}$ can involve various tools such as SQL executor or search engine, we utilize Python due to its extensive mathematical libraries and programming flexibility.

To evaluate the quality of generated data analysis $\mathbf{y}$, we use **helpfulness** as the primary metric. Motivated by literature in the data analysis field [22], we define helpfulness as: (1) relevance to the query, (2) effective and insightful data interpretation, and (3) diversity in terms of analysis perspectives. We evaluate helpfulness through **pairwise comparison** following common approach [29, 38, 45]. Given two analyses generated by two different systems, the annotator (either human or simulated by LLM) selects the more helpful one based on our defined criteria. The winning rate of each system is reported as helpfulness score. To obtain a comparable set of numbers for all models, we report the winning rate of each model against the annotations, so a score of 50 would indicate the model generations are perceived as helpful as annotations.

## 3 Dataset Construction

We construct our DACO dataset through four stages: (1) database collection, (2) query collection, (3) automatic annotation collection, and (4) human refinement. The workflow is shown in Figure 2(a).

**Database collection.** We collect databases from two sources: Spider [42] and Kaggle (`https://www.kaggle.com/datasets`). There are 157 databases collected from Spider, which originally come from university databases, DatabaseAnswers and Wikipedia. We then crawl and filter 5,830 databases from Kaggle. Despite the initial filtering, the quality of these databases remains low, often containing noisy or unintelligible data. Therefore, we conduct a manual inspection and thereby select a subset of 314 clean and interpretable databases to build our dataset. To maintain the diversity of the resulting database set, 157 of the databases are deliberately chosen from the long tail of the topic distribution. We employ BERTopic [11] to model the topic distribution, which produces in total 160 topics. We take its least frequent 80 topics as the long tail, which covers 26.79% of the total databases.

**Query collection.** We generate 10 queries for each database by prompting ChatGPT to first assume the role of a database stakeholder and then generate an application-driven query based on the role. To ensure the quality of the query, we perform a manual filtering to the machine generated queries. Specifically, we remove queries that are not driven by real-world applications or cannot be answered by the given reference database. We train a group of 6 annotators to perform such a filtering process. As a result, there are about 42% of the queries removed, where the removal agreement achieves a 0.62 cohen kappa score. After the aforementioned processes, we obtain in total 2,664 queries. Examples of databases and the automatically generated queries are shown in Figure 8.

**Automatic annotation collection.** As shown in Figure 2(b), we design a pipeline that leverages the code generation capability of LLMs to automate the answer annotation for our DACO dataset. Based on the database and the query, we instruct the LLM to perform data analysis in multiple turns. At each turn, the LLM will produce a python code snippet and take its execution outputs as evidences to reason over and support its follow-up interpretation. After each turn, we prompt the model to decide whether the analysis is sufficiently comprehensive; if deemed sufficient, it terminates the coding turns and produces the final answer. With this pipeline, we instruct GPT-4 to automatically generate all the answer annotations to each query of our dataset, for both the intermediate code and the final analysis answering the queries. To improve the quality of such automatically constructed annotations, we additionally allow GPT-4 to correct its own mistakes when its generated code leads to run-time or syntax error, where only the corrected codes are kept. In total, we obtain 1.9k valid query-answer pairs, each with roughly 3.3 intermediate coding steps.

**Human refinement.** The annotated analyses thus far have been algorithmically generated, where their actual quality are to be further verified. We thus curate a human-refined subset containing 100 densely human-annotated query-answer pairs as illustrated in Figure 2(c). For each query, we sample

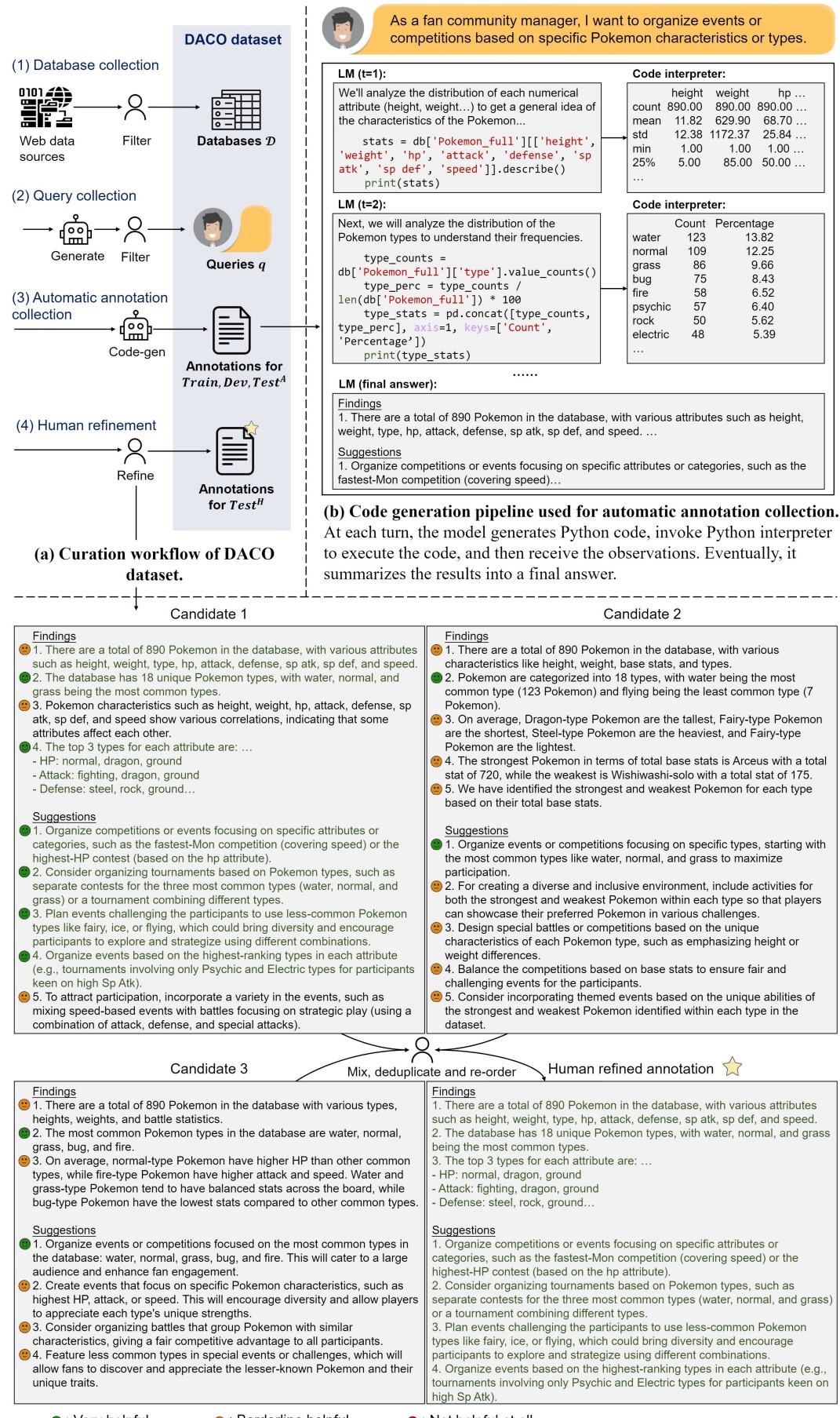

**(b) Code generation pipeline used for automatic annotation collection.** At each turn, the model generates Python code, invoke Python interpreter to execute the code, and then receive the observations. Eventually, it summarizes the results into a final answer.

**(a) Curation workflow of DACO dataset.**

🟢 : Very helpful          🟠 : Borderline helpful          🔴 : Not helpful at all

Green highlighted text: bullet points that are incorporated into the final human refined annotation

**(c) Human refinement.** For each query, we sample 3 candidate analysis to be refined by human annotators. The annotators are asked to rate the helpfulness of each bullet point into three levels (*very helpful*, *borderline helpful*, or *not helpful at all*) based on our definition of helpfulness, and then mix the high-quality bullet points into the final annotation with deduplication and reordering. In this example, all bullet points in the human annotations are from Candidate 1, because the very helpful points from Candidate 2 are 3 are mostly duplicate of those in Candidate 1.

Figure 2: An overview of DACO construction.

Table 1: **Statistics of DACO dataset. Left:** We report the size of each data split, including the number of databases (# db), the number of queries (# queries), the number of bullet points (# bullets) and tokens (# tokens) in output answer $\mathbf{y}$, and the number of code steps and code lines in intermediate coding steps $\mathbf{a}$. Train, Dev and Test$^A$ sets are automatically generated with GPT-4, while Test$^H$ is the human refined subset and only contains the final output answer. **Right:** We report more fine-grained statistics regarding the input databases and output answers. For all databases in DACO, we report the number of tables, columns and rows within each database. The numbers of rows and columns for each database are calculated by summing across all tables within the database. For output answers in Test$^H$, we report the number of findings, suggestions and tokens in each answer.

| | Train | Dev | Test$^A$ | Test$^H$ | Total | | Med. | Max | Min |
|---|---|---|---|---|---|---|---|---|---|
| *Input Statistics* | | | | | | *Detailed statistics for db* | | | |
| # db | 353 | 22 | 65 | 17 | 440 | # tables | 1 | 15 | 1 |
| # queries | 1558 | 100 | 284 | 100 | 1942 | # columns | 15 | 96 | 3 |
| *Annotation Statistics* | | | | | | # rows | 572 | 67.2$k$ | 4 |
| # bullets | 14.8$k$ | 996 | 2728 | 980 | 19.5$k$ | *Detailed statistics for answer* | | | |
| # tokens | 575$k$ | 36.6$k$ | 106$k$ | 42.3$k$ | 760$k$ | # findings | 5 | 8 | 3 |
| # code steps | 5086 | 346 | 948 | - | 6380 | # suggestions | 5 | 8 | 3 |
| # code lines | 3.0$M$ | 208$k$ | 555$k$ | - | 3.7$M$ | # tokens | 397 | 864 | 202 |

3 different analysis candidates using the previously described automated method with GPT-4. We ask the annotators to evaluate the quality of each machine generated bullet point and categorize each point into one of *not helpful*, *borderline helpful*, or *very helpful*. The highest quality points from the three candidates were combined and refined into a final gold-standard analysis. Concretely, the annotators should first combine all *very helpful* points, remove duplicate points, reorder the points to maintain coherence, and make necessary textual edits for fluency. Suppose the number of bullet points are lower than our pre-defined lowest threshold (3 bullet points per answer), the annotators should select additional bullet points ranked as *borderline helpful* to augment the answer. we ask a group of 3 internal members to perform refinement. The agreement accuracy of the refinement process (candidate point selection) is 0.83 and the Cohen's Kappa is 0.67.

## 4 Data Statistics

DACO dataset includes training, development and test sets with annotations generated by GPT-4. Furthermore, DACO features a human-refined testing subset. To differentiate the two test sets, we denote the automatically annotated set as Test$^A$ and the human-refined set as Test$^H$. Detailed statistics are in Table 1.

**Database statistics.** In total, DACO comprises 440 databases, each of which contains on average 2.3 tables. To better visualize the *major topic* distribution of this selected subset, we again use BERTopic but group these databases into 10 topics. The keywords for top 5 topics are shown in Figure 3. The leading topic (topic 1) is associated with business setting and consists of 46.52% of the dataset. The remaining nine topics exhibit a relatively even distribution, covering a broad range of domains, including sports (topic 2), healthcare (topic 3), weather (topic 4), and education (topic 5).

**Query statistics.** In average, each database is accompanied by 4.4 different user queries. We show the top 15 verbs and their top 3 direct noun objectives in Figure 4. The queries demonstrate a notable level of diversity. The most common type of queries is to request analysis (such as "analyze data" and "identify pattern"), followed by queries aiming to make decisions (such as "determine strategy" and "make decision"). To quantatively verify the diversity of input queries, we measure the overlap between queries over the same database using cosine similarity of Sentence-BERT [32] embeddings. Based on manual inspection of various cases, we categorize the overlap between query pairs into *low*, *medium*, and *high* at thresholds of 0.5 and 0.8. Representative examples are shown in Table 2. We find that 45.6% of generated query pairs have low similarity, 52.4% have medium similarity, and only 2.0% are highly similar. The small percentage of highly similar pairs suggests high diversity of input queries.

**Annotation statistics.** Figure 5 shows the distribution of the top 10 API functions invoked in the generated code. Excluding the trivial `print` function, the most frequent APIs are `pandas` APIs including table manipulation (e.g. `groupby` and `merge`) and mathematical computation (e.g. `mean`

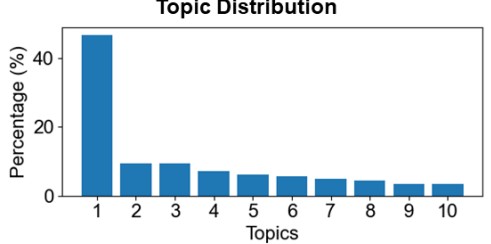

Figure 3: **Domain distribution of databases in** D**ACO** **dataset.** Upper section: distribution of the 10 topics of databases, showing a long-tail effect. Lower section: keywords of the top five topics explaining the topics' content.

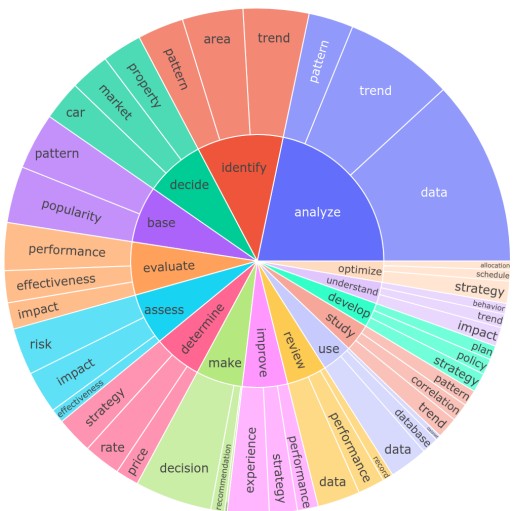

Figure 4: **Distribution of queries in** D**ACO** **dataset.** We display the top 15 verbs and their top 3 direct noun objectives, demonstrating the query diversity.

Table 2: Percentage and representative examples of query pairs with low, medium and high overlap. In the representative examples, repetitive information and distinctive information are highlighted.

| Overlap | % | Representative Examples |
|---|---|---|
| Low ($sim < 0.5$) | 45.6 | *Query 1:* As an advertising executive, I want to select the channels for targeted ad placements. 
 *Query 2:* As a platform developer, I want to analyze user behavior and preferences to optimize the user experience. |
| Medium ($0.5 < sim < 0.8$) | 52.4 | *Query 1:* As a farmer, I want to determine the suitable fruit varieties to grow on my farm. 
 *Query 2:* As a fruit exporter, I want to identify the fruits that meet export standards and have a longer shelf life. |
| High ($sim > 0.8$) | 2.0 | *Query 1:* As a consultant for honey market, I want to study the honey production trend to recommend business strategies for my clients. 
 *Query 2:* As a curious analyst, I want to study the production trend to understand the US honey industry. |

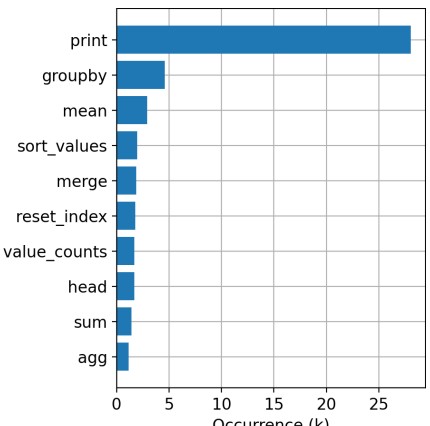

Figure 5: **Top 10 API functions invoked in the generated code.**

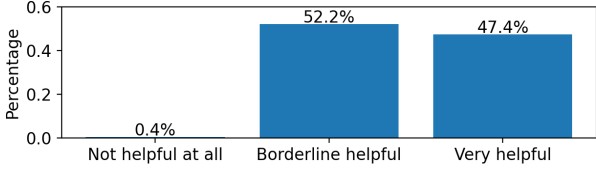

Figure 6: **Helpfulness of Test$^A$ annotations evaluated by humans.**

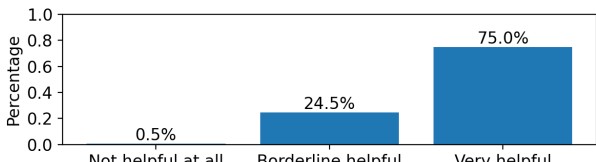

Figure 7: **Helpfulness of Test$^H$ annotations evaluated by humans.**

and `sort_values`). Figure 6 and 7 displays the human rating for each bullet points in Test$^A$ and Test$^H$ collected during the human refinement stage. While automatic annotations significantly lag behind manually refined ones, they still offer valuable knowledge that can be leveraged through supervised fine-tuning, as demonstrated in the experiments section. For manually refined annotations, mentioned in Section 3, the agreement accuracy is 0.83 and Cohen's Kappa is 0.67, verifying their quality. An example of human refinement is shown in Figure 2.

## 5 Experiments

### 5.1 Experimental Settings

**Table QA baselines.** We experiment with two models designed for table QA tasks, TAPAS [13] and TAPEX [19]. TAPAS is a BERT-style model pre-trained to select relevant information from a table based on user query. For our dataset, we first use TAPAS to select relevant information and then use ChatGPT to interpret the selected information. TAPEX is a encoder-to-decoder model pre-trained on table and SQL data. We fine-tune TAPEX with GPT-4-generated annotations.

**Prompt-based LLMs.** We evaluate the performance of ChatGPT and GPT-4 when enhanced by code generation. For comparison, we also experiment with a baseline counterpart that does not include code generation and instead directly takes raw table content as input.

**Fine-tuned LLMs.** With the answer annotation generated by GPT-4, we train a 6B CodeGeeX2-6B [44] model through SFT, RLHF, and fine-grained RLHF (denoted as FG-RLHF) [38].

We begin by training the model with SFT, using either all annotations including code generation or purely the final answer annotations. The former SFT model is further refined through RLHF with an end goal of optimizing the final answer $\mathbf{y}$'s helpfulness. We model the helpness of final answer with an *answer RM $R_a$*. Concretely, $R_a$ is trained on pairwise comparison data of output bullet points annotated by ChatGPT. To encourage diversity, we additionally impose repetition penalty by subtracting a similarity score between different bullet points from the final reward.

However, we observe that RLHF struggles to provide useful supervision signal to intermediate code generation since its reward is only applied to the final answer. Recent work has shown that providing dense reward for the intermediate steps can alleviate this issue [18, 38]. Thus, we propose to use fine-grained RLHF [38], where we introduce two novel reward models to provide dense reward signals for code generation: *contribution RM $R_c$* and *regularization RM $R_r$*. $R_c$ encourages code generation that better contributes to the final answer at each step, while $R_r$ helps prevent reward hacking. Concretely, we compute the similarity $Sim(\mathbf{y}, \mathbf{o}_i)$ between final answer $\mathbf{y}$ and the $i$-th step observation $\mathbf{o}_i$ to measure the contribution of generated code action $\mathbf{a}_i$. Using $Sim(\mathbf{y}, \mathbf{o}_i)$, we can rank the contribution of different steps, and then use the comparison pairs $(\mathbf{a}_i, \mathbf{a}_j)$ to train $R_c$. However, this heuristic training of $R_c$ may lead to the well-studied reward misspecification issue termed reward hacking [34]. To mitigate this issue, we propose to regularize such behavior with $R_r$. Given the misspecified reward model $R_c$, we first train an RL model until its generations start to collapse to certain patterns, then we train $R_r$ to detect such patterns and thus assign lower scores.

**Evaluation.** The main metric we use is **pairwise comparison of helpfulness** as in Section 2. We use three LLM evaluators, GPT-4o mini, Claude 3.5 Sonnet, and Llama 3 8B [2], as well as trained human annotators for the evaluation. We additionally report BLEU score, entailment score, and helpfulness evaluation for each individual bullet point. These metrics cannot holistically measure the analysis helpfulness, but can provide complementary insights for analyzing model performance. For entailment, we use an off-the-shelf NLI model to compute the probability that the model generation is entailed by the annotation. For point-wise evaluation, we ask the annotator to assign a score chosen from 0 (*not helpful*), 1 (*borderline helpful*) and 2 (*very helpful*) to each bullet point using the same standard as in human refinement of test set. Our human annotation achieves high agreement of 0.62 Cohen's kappa for pairwise comparison of helpfulness, and 0.65 Cohen's kappa for point-wise helpfulness evaluation.

### 5.2 Results

The main results are in Table 3. The key conclusions are as follows:

Table 3: **Main results.** We report helpfulness (Help.), entailment (Entail.), and BLEU on both automatically annotated test set (Test$^A$) and human curated test set (Test$^H$). The helpfulness score is the average of scores evaluated by GPT-4o mini, Claude 3.5 Sonnet, and Llama 3 8B. Highest numbers in each section are highlighted in bold. We also report the number of parameters (# para.) of each model. †: For ChatGPT and GPT-4, we report the number of parameters based on our best estimation.

|  | Method | # para. | Code gen | Test$^A$ | | | Test$^H$ | | |
|---|---|---|---|---|---|---|---|---|---|
|  |  |  |  | Help. | Entail. | BLEU | Help. | Entail. | BLEU |
| *TableQA* | TAPAS | 337M | ✗ | **19.19** | 1.96 | 11.62 | **16.50** | **3.67** | 9.73 |
| *Baselines* | TAPEX | 406M | ✗ | 15.08 | **3.34** | **14.60** | 9.00 | 3.50 | **13.81** |
| *Prompt-* | ChatGPT | 20B$^†$ | ✗ | 19.31 | 3.06 | 13.22 | 13.50 | 2.07 | 13.51 |
| *based LLMs* | GPT-4 | 175B$^†$ | ✗ | 30.43 | 3.35 | 14.90 | 20.50 | **4.36** | 13.71 |
|  | ChatGPT | 20B$^†$ | ✓ | 26.51 | 2.74 | 14.22 | 21.38 | 2.59 | 14.51 |
|  | GPT-4 | 175B$^†$ | ✓ | **50.79** | **4.59** | **17.77** | **43.92** | 3.26 | **17.54** |
| *Finetuned* | SFT | 6B | ✗ | 18.96 | 2.30 | 14.47 | 11.33 | 2.65 | 13.63 |
| *LLMs* | SFT | 6B | ✓ | 13.73 | 2.15 | **14.88** | 9.83 | 4.47 | **14.60** |
|  | RLHF | 6B | ✓ | 10.64 | 3.18 | 12.66 | 7.51 | 3.13 | 11.46 |
|  | FG-RLHF | 6B | ✓ | **19.42** | **3.65** | 13.13 | **12.50** | **5.98** | 11.80 |

Table 4: **Human evaluation.** We report human-rated and LLM-rated helpfulness pairwise comparison of two pairs of models: GPT-4 with v.s. without code generation, and FG-RLHF v.s. SFT. We also report point-wise evaluation scores scaled into $0 \sim 2$ rated by human annotators.

|  | Pairwise comparison | | Point- |
|---|---|---|---|
|  | Human | LLM | wise |
| GPT-4 code gen *v.s.* | **66.41** | **70.07** | **1.45** |
| GPT-4 w/o code gen | 33.59 | 29.93 | 1.36 |
| FG-RLHF *v.s.* | **57.72** | **58.49** | **1.42** |
| SFT | 42.28 | 41.51 | 1.30 |

Table 5: **APIs** ranked by its correlation with contribution RM scores. Higher correlation means that contribution RM assigns higher scores to code snippets containing the API.

| Top 4 APIs | | Bottom 4 APIs | |
|---|---|---|---|
| API | Corr. | API | Corr. |
| print | 44.24 | to_datetime | -18.96 |
| nlargest | 20.06 | isnull | -17.76 |
| mean | 14.56 | describe | -12.02 |
| sort_values | 12.23 | merge | -10.83 |

**Proprietary LLMs demonstrate strong data analysis capabilities but still lag behind human performance.** Proprietary LLMs, especially GPT-4, perform the best and significantly outperform other models on almost all metrics, particularly when enhanced with code generation. However, pairwise comparisons between model generation and human-refined annotations (Test$^H$) show that even the best model only wins 41.88% of the time, indicating a gap in generating helpful data analysis.

**The fine-tuned models show promising data analysis capabilities, demonstrating the usefulness of our weak supervision data.** By training with automatically generated annotations, SFT with code generation achieves a reasonable helpfulness score and outperforms the TAPEX baseline. Though RLHF negatively affects performance due to sparsity of reward signals, FG-RLHF with our dense reward models significantly improves the performance, outperforming SFT by 7 points in helpfulness. Despite the difference in model size, FG-RLHF outperforms ChatGPT without code generation in helpfulness and entailment metrics. Human evaluation shows a 57.72% win rate of FG-RLHF over SFT, as seen in Table 4. Our qualitative analysis indicates that FG-RLHF better focuses on user queries, while SFT tends to display generic statistics less relevant to user queries.

**The fine-tuned models show promising data analysis capabilities, demonstrating the usefulness of our weak supervision data.** By training with automatically generated annotations, SFT with code generation achieves a reasonable helpfulness score and outperforms the TAPEX baseline. Though RLHF negatively affect the performance due to the sparsity of reward signal, we see that FG-RLHF with our designed dense reward models significantly boosts the generation helpfulness, and outperforms SFT by 7 points on helpfulness, thus better aligning with human preference. Despite the difference in model size, FG-RLHF outperforms ChatGPT w/o code generation on helpfulness and entailment metrics. Human evaluation demonstrates a 57.72 win rate of FG-RLHF over SFT as in Table 4. Our qualitative analysis shows that FG-RLHF better focuses on user query, while SFT tends to display generic statistics that are less relevant to user query.

We analyze the behavior of two reward models to better understand their effects on FG-RLHF. The *contribution RM* favors API calls that extract important information from tabular data but is vulnerable

to reward hacking. As in Table 5, the functions rewarded most are related to extracting significant features (`nlargest`, `sort_values`), aggregating results (`mean`), and displaying specific information (`print`). In contrast, the least rewarded functions involve displaying generic statistics (`describe`) and wrangling data (`merge`, `to_datetime`, `is_null`) since they cannot directly contribute to the user query. However, the concerningly high correlation between `print` function and contribution RM scores indicates the policy may exploit the correlation to hack reward, which can be mitigated by employing the regularization RM.

**Evaluation on external test sets.** We further evaluate our fine-tuned models on two external test sets: (1) InfiAgent-DA benchmark that focuses on complex but not application-driven data analysis [14], and (2) free-form table question answering dataset FeTaQA [26]. We find that FG-RLHF improves the accuracy over SFT on InfiAgent-DA (14.61 v.s. 12.92), especially over questions about summary statistics (14.86 v.s. 10.80) and correlation analysis (21.57 v.s. 14.86), which aligns with our evaluation results on FG-RLHF dataset. On FeTaQA, FG-RLHF retains similar performance (6.35 Rouge-L, 80.74 BERTScore) compared to SFT (6.39 Rouge-L, 80.68 BERTScore) since FG-RLHF is not specifically trained to enhance information lookup capabilities.

# 6   Related Work

**Table Analysis.** Early work in table question answering (table QA) targets simple questions that requires table lookup and cell aggregations [31, 46, 16, 42, 26]. Later benchmarks further require free-form answer generation [26], multi-hop reasoning [7, 8] and mathematical reasoning [47, 9, 24]. Despite the similar formulation between our task and existing table QA work, their focus are different: most existing table QA datasets focus on obtaining specific information, our data analysis queries can be complex and requires query decomposition and reasoning. Some concurrent work further targets comprehensive table analysis such as correlation analysis and causal reasoning [27, 14, 20]. The main difference between this work to the concurrent work is our focus on addressing application-driven and complex user queries, which requires more reasoning skills.

**Code Generation.** Code generation benchmarks have been proposed for general-purpose programming [3, 12], math problems [3], and data science scenario [17, 15]. Similar to our work, some recent work allows the language model to interact with a code execution environment and receive execution outputs as feedback [39, 37]. The most relevant work is [10] that also addresses data analysis via code generation. Given a data analysis query, they use GPT-4 to first generate code and then provide an interpretation of the execution results. While their analysis queries are still relatively simple, this is an early exploration aiming at automating data analysis.

**LLM Agent.** The notable capability of LLMs in reasoning and planning inspires many work to use them as intelligent *agents* for complex tasks like Minecraft gaming [36], robotics control [1], and web browsing [41]. In this work, our code generation pipeline can be considered as an adaptation of ReAct [40], where the LLM iteratively generates code as actions and reads execution results as observations. Although we have not yet explored more advanced agent designs, such as tool-crafting agents [6, 43] or self-reflection agents [33, 25], these approaches could be adapted to our task by redefining the action space as code generation and the observation space as execution results. We leave this exploration for future work.

# 7   Conclusion

In this work, we propose a novel and challenging data analysis task, which involves decomposing user query into multiple perspectives, grounding each perspective to the input data and performing logical and mathematical reasoning. To support this task, we build the DACO dataset containing large-scale annotations automatically generated by GPT-4 and a small but high-quality test set with human curated annotations. We evaluate three types models on our dataset: table QA models, prompt-based proprietary LLMs, and open LLMs fine-tuned on our automatically collected annotations. While GPT-4 consistently performs the best, the fine-tuned models achieves reasonably good helpfulness with much less computation. On top of the SFT model, we further show that fine-grained RLHF can be employed to boost helpfulness perceived by humans.

## Acknowledgement

This work was partially supported by ByteDance during Xueqing's internship with the company. This work was also supported partially by an Amazon gift grant.

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

## A  Appendix

Project website is at `https://shirley-wu.github.io/daco/index.html`. Data and code are released at `https://github.com/shirley-wu/daco`. Croissant metadata record is at `https://github.com/shirley-wu/daco/blob/main/data/croissant.json`. We license our resources under Apache-2.0 license.

We thereby state that we bear all responsibility in case of violation of rights, etc., and confirmation of the data license.

## B Dataset Documentation

Below are dataset documentation following the framework from datasheets for datasets:

**Motivation:**

- *For what purpose was the dataset created?* - For the novel task of data analysis as explained in the main content.
- *Who created the dataset and on behalf of which entity?* - This dataset is created during a collaboration of ByteDance AI Lab and University of California, Los Angeles.
- *Who funded the creation of the dataset?* - ByteDance

**Composition:**

- *What do the instances that comprise the dataset represent?* - Each instance contains a database of tabular data, a question, a reasoning process including code snippets, and a final answer. Everything is in text format, except the database is stored as `pd.DataFrame`
- *How many instances are there in total?* - As detailed in Table 1 in the main content.
- *Does the dataset contain all possible instances or is it a sample of instances from a larger set?* - The dataset is not a sample from a larger set.
- *What data does each instance consist of?* - Raw data.
- *Is there a label or target associated with each instance?* - Yes, as explained in Section 3 in the main content.
- *Is any information missing from individual instances?* - No
- *Are relationships between individual instances made explicit?* - N/A
- *Are there recommended data splits?* - Yes. We split the dataset randomly, and encourage people to follow this split for reproductivity. We also curate human annotations only for the test set.
- **Are there any errors, sources of noise, or redundancies in the dataset?** - Yes. The input questions and answer annotations are generated by ChatGPT, which will inevitably contain errors. We try to manage the affect by manually filtering the questions, and by curating a test set of human refined answer annotations.
- *Is the dataset self-contained, or does it link to or otherwise rely on external resources?* - Self-contained.
- *Does the dataset contain data that might be considered confidential?* - No.
- *Does the dataset contain data that, if viewed directly, might be offensive, insulting, threatening, or might otherwise cause anxiety?* - No.
- *Does the dataset identify any subpopulations?* - No.
- *Is it possible to identify individuals, either directly or indirectly from the dataset?* - No.
- *Does the dataset contain data that might be considered sensitive in any way?* - No.

**Collection process:** as described in Section 3 in the main content.

**Preprocessing/cleaning/labeling**: we release the raw text data and do not perform any preprocessing/cleaning/labeling of the texts.

**Uses:**

- *Has the dataset been used for any tasks already?* - The data analysis task, as in the main content
- *Is there a repository that links to any or all papers or systems that use the dataset?* - `https://github.com/shirley-wu/daco`

- *What (other) tasks could the dataset be used for?* - As in the main content

- *Is there anything about the composition of the dataset or the way it was collected and preprocessed/cleaned/labeled that might impact future uses?* - N/A

- *Are there tasks for which the dataset should not be used?* - N/A

**Distribution:**

- *Will the dataset be distributed to third parties outside of the entity on behalf of which the dataset was created?* - Yes

- *How will the dataset will be distributed?* - `https://github.com/shirley-wu/daco`

- *When will the dataset be distributed?* - Already released

- *Will the dataset be distributed under a copyright or other intellectual property (IP) license, and/or under applicable terms of use (ToU)?* - Yes, Apache-2.0 license

- *Have any third parties imposed IP-based or other restrictions on the data associated with the instances?* - No

- *Do any export controls or other regulatory restrictions apply to the dataset or to individual instances?* - No

**Maintanance:**

- *Who will be supporting/hosting/maintaining the dataset?* - The dataset is not planned to be a dynamic dataset, but the authors will keep maintaining the github repo

- *How can the owner/curator/manager of the dataset be contacted?* - Github or email `xueqing.wu@cs.ucla.edu`

- *Is there an erratum?* - No

- *Will the dataset be updated?* - No unless to correct errors

- *Will older versions of the dataset continue to be supported/hosted/maintained?* - N/A

- *If others want to extend/augment/build on/contribute to the dataset, is there a mechanism for them to do so?* - Yes, please feel free to do that as long as citing our work and following the license

## C   Limitations

While we put forth to construct the first of its kind dataset, DACO, for the comprehensive data analysis task, the challenging nature of the data analysis process (which often requires certain domain expertise) itself presents two major limitations of this work: **(1) It is expensive to build expert-annotated dataset.** In our work, the large-scale annotations are automatically generated by GPT-4, and their quality cannot always be well guaranteed. Although we curate a test set refined by humans, those answers are initially generated by GPT-4, which may introduce biases to human annotators during the refinement process for the final answer. Annotations curated by experts from scratch might have higher quality, but they are indeed quite costly and can create other sort of alignment problems. **(2) It is nontrivial to evaluate model generations.** Evaluating the quality of the analyses is by itself challenging and requires data science expertise. For automatic evaluation, we use ChatGPT to rank the helpfulness of model generations, which can partially but not perfectly align with human preference. Additionally, ChatGPT cannot always robustly evaluate the correctness of model generations. We use an off-the-shelf NLI model to evaluate the entailment probability between human-refined ground truths and the model generations, which can partially reflect the correctness of model generations. However, the entailment probability prediction can sometime propagate errors which lead to false positives or negatives. We make efforts to alleviate such an issue by additionally collecting human evaluations, which are supposed to better reflect the answer quality, despite that humans can occasionally exhibit subjective evaluation patterns. Notice that our annotators do not fully check the correctness of the generated answers, where we task them to focus more on the helpfulness metrics defined in this work.

# D   Implementation Details

For **zero-shot API-based systems** including ChatGPT and GPT-4, we evaluate two settings, directly reading the table content, and using code generation. For the former setting, we linearize the table content into text representation as model input. Due to token limit, we feed the first 20 rows as input, which covers the full content of 93% tables. For the code generation setting, we employ the pipeline described in Figure 2(b) in the main content. When the generated code causes a syntax or runtime error, we re-sample the model until the generated code can be executed. We allow up to 5 resamplings for each turn. We use the `gpt-3.5-turbo-16k-0613` API for ChatGPT and `gpt-4-32k` API for GPT-4. We limit the number of total coding turns maximally at 9. For annotation generation where GPT-4 self-correction is allowed, we limit the number of self-correction within 2 for each turn and 4 for the whole session.

For **finetuned models** including SFT, RLHF and fine-grained RLHF, we use CodeGeeX2-6B [44] as the base model. We first train the SFT model using GPT-4 annotations, and then train our RLHF models on top of the SFT model. When training $R_{a+c}$ and $R_r$, we initialize the model from the SFT model. When training our fine-grained RLHF model, we initialize the value model $V$ from $R_{a+c}$, and initialize the policy model $\pi$ from the SFT model. In inference, we use nucleus decoding with $p = 0.9$ and temperature = 1.0. Similarly, we allow up to 5 resamplings when the generated code causes an error. The SFT model is trained with 8 A100 GPU for about 4 hours. The RLHF models are trained with 8 A100 GPU for about 18 hours. Detailed hyper-parameters are in Table 6. The only hyper-parameter we tune is $\lambda$ for fine-grained RLHF. We experiment with 0.8, 0.9 and 1.0 and discover that 1.0 works the best.

|                      | SFT  | RL   |
| -------------------- | ---- | ---- |
| learning rate        | 1e-5 | 2e-6 |
| gradient accumulation | 4    | 4    |
| total steps          | 600  | 200  |
| $\lambda$            | -    | 1.0  |
| $\gamma$             | -    | 1.0  |

Table 6: **Hyperparameters.**

# E   Details of LLM Evaluation

We adopt LLM for automatic evaluation due to the complexity of evaluating answer helpfulness. Prior work has shown that LLMs can reliably evaluate text generation quality based on task description and judging criteria [21]. However, it has been known that LLM evaluators favor their own generations [30]. To mitigate this potential bias, we evaluate model helpfulness with multiple LLM evaluators, including OpenAI's `gpt-3.5-turbo-0613` (now deprecated) and `gpt-4o-mini-2024-07-18`, Anthropic's `claude-3-5-sonnet-20240620`, and `meta-llama/Meta-Llama-3-8B-Instruct` from Llama 3 family. The detailed evaluation results are in Table 7. We report the average scores from GPT-4o mini, Claude 3.5 Sonnet and Llama 3 8B as the final score. Additionally, we show that different LLM evaluators agree with each other, achieving a moderate to substantial inter-annotator agreement of 0.45 Cohen's kappa and a very high Spearman correlation of 0.90 in model performance ranking. This further verifies the robustness of LLM-based evaluation.

# F   License Information

Based on Kaggle's policy, it is allowed to use and redistribute datasets as long as adhering to each dataset's specific license. All datasets utilized in this study are publicly accessible. A majority, 65%, are covered under various standardized licenses, all of which permit data redistribution for academic purposes. All these licenses allow data redistribution for academic purposes. The remaining 35% are licensed on an ad-hoc basis. We have manually reviewed their Kaggle descriptions to ensure there are no restrictions against using their data. However, since some datasets have limited description or licensing information, we will continue to monitor their status and will remove any dataset if issues arise. Detailed license information is listed in Table 8.

Table 7: **LLM evaluation results.** We report the helpfulness score on Test$^A$ and Test$^H$ evaluated by four different LLM evaluators, GPT-3.5 Turbo (now deprecated), GPT-4o mini, Claude 3.5 Sonnet and Llama 3 8B.

| | Code gen | Test$^A$ | | | | Test$^H$ | | | |
|---|---|---|---|---|---|---|---|---|---|
| | | GPT-3.5 | GPT-4o | Claude 3.5 | Llama 3 | GPT-3.5 | GPT-4o | Claude 3.5 | Llama 3 |
| TAPAS | ✗ | 25.00 | 17.61 | 17.08 | 22.89 | 24.50 | 15.00 | 15.00 | 19.50 |
| TAPEX | ✗ | 14.79 | 17.78 | 7.39 | 20.07 | 6.00 | 9.50 | 3.50 | 14.00 |
| ChatGPT | ✗ | 25.18 | 20.60 | 18.31 | 19.01 | 18.50 | 16.00 | 9.00 | 15.50 |
| GPT-4 | ✗ | 30.81 | 30.81 | 32.39 | 28.17 | 24.00 | 21.00 | 23.00 | 17.50 |
| ChatGPT | ✓ | 35.74 | 22.89 | 24.70 | 31.93 | 27.27 | 18.69 | 19.19 | 26.26 |
| GPT-4 | ✓ | 52.00 | 51.09 | 52.18 | 49.09 | 41.88 | 45.31 | 39.58 | 46.88 |
| SFT | ✗ | 21.51 | 19.72 | 12.68 | 24.47 | 9.50 | 11.50 | 5.00 | 17.50 |
| SFT | ✓ | 20.95 | 12.38 | 7.38 | 21.43 | 11.54 | 5.76 | 7.05 | 16.67 |
| RLHF | ✓ | 15.18 | 14.51 | 3.13 | 14.29 | 8.79 | 7.14 | 0.55 | 14.84 |
| FG-RLHF | ✓ | 28.54 | 20.71 | 9.51 | 28.05 | 21.05 | 13.82 | 7.24 | 16.45 |

| License | Count |
|---|---|
| CC0-1.0 | 123 |
| Ad-hoc | 110 |
| DbCL-1.0 | 14 |
| Attribution 4.0 International (CC BY 4.0) | 14 |
| CC-BY-NC-SA-4.0 | 13 |
| CC-BY-SA-4.0 | 7 |
| ODbL-1.0 | 6 |
| GPL-2.0 | 4 |
| CC BY 4.0 | 4 |
| Community Data License Agreement - Permissive - Version 1.0 | 3 |
| Attribution-NonCommercial 4.0 International (CC BY-NC 4.0) | 3 |
| MIT License | 2 |
| Open Government Licence v3.0 | 2 |
| ODC Public Domain Dedication and Licence (PDDL) | 2 |
| ODbL | 1 |
| UN Data License | 1 |
| ODC Attribution License (ODC-By) | 1 |
| Creative Commons Attribution 4.0 International License | 1 |
| Open Use of Data Agreement v1.0 | 1 |
| Attribution-NonCommercial-ShareAlike 3.0 IGO (CC BY-NC-SA 3.0 IGO) | 1 |
| Government Open Data License - India | 1 |

Table 8: License information for Kaggle datasets used in our work.

# G   Qualitative Examples

We show a few examples of input databases and their associated queries in Figure 8. Though the queries are automatically generated, they are of high quality, diverse, and relevant to the databases.

We show final answers generated by SFT and RLHF in Figure 9. RLHF better focuses on user query, while SFT tends to display generic statistics that are less relevant to user query.

We show examples of code generations in Figure 10. We also report their reward scores from contribution RM and regularization RM.

# H   Prompts

Prompt for query generation is in Table 9. Prompt for helpfulness annotation collection is Table 10. Prompt for helpfulness evaluation is Table 11.

**Database: course teach**

course

| Course_ID | Staring_Date | Course |
|---|---|---|
| 1 | 5 May | Language Arts |
| 2 | 6 May | Math |
| 3 | 7 May | Science |
| 4 | 9 May | History |
| … | … | … |

course_arrange

| Course_ID | Teacher_ID | Grade |
|---|---|---|
| 2 | 5 | 1 |
| 2 | 3 | 3 |
| 3 | 2 | 5 |
| 4 | 6 | 7 |
| … | … | … |

teacher

| Teacher_ID | Name | Age | Hometown |
|---|---|---|---|
| 1 | Joseph Huts | 32 | Blackrod Urban District |
| 2 | Gustaaf Deloor | 29 | Bolton County Borough |
| 3 | Vicente Carretero | 26 | Farnworth Municipal Borough |
| 4 | John Deloor | 33 | Horwich Urban District |
| … | … | … | … |

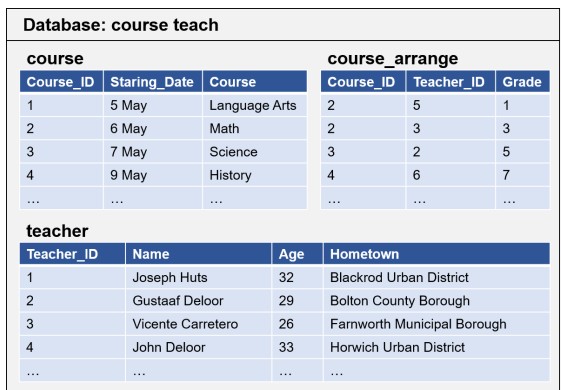

Questions:
- As the head of curriculum development, I want to analyze course offerings and make recommendations for changes or improvements.
- As a teacher union representative, I want to review course schedules and teacher assignments to ensure fair workload distribution.
- As a budget analyst, I want to analyze course offerings and course enrollment to determine budget allocation and resource allocation.
- As a school counselor, I want to review course offerings and enrollment data to provide guidance and recommendations for college and career planning for students.
- As a school registrar, I want to review course offerings and enrollment data to ensure accurate records and transcripts for students.

(a) Example 1.

**Database: SATELLITES AND DEBRIS IN EARTH'S ORBIT**

space_decay

| CREATION_DATE | OBJECT_NAME | OBJECT_ID | OBJECT_TYPE | CENTER_NAME | MEAN_MOTION | ECCENTRICITY | INCLINATION | … |
|---|---|---|---|---|---|---|---|---|
| 2021-11-01T06:46:11 | ARIANE 42P+ DEB | 1992-072J | DEBRIS | EARTH | 2.921700 | 0.652893 | 7.7156 | … |
| 2021-11-01T04:58:37 | SL-8 DEB | 1979-028C | DEBRIS | EARTH | 13.754973 | 0.003072 | 82.9193 | … |
| 2021-11-01T06:26:11 | GSAT 1 | 2001-015A | PAYLOAD | EARTH | 1.038224 | 0.023739 | 12.1717 | … |
| 2021-10-31T18:07:15 | CZ-4 DEB | 1999-057MB | DEBRIS | EARTH | 14.775907 | 0.006062 | 98.4781 | … |
| … | … | … | … | … | … | … | … | … |

Questions:
- As a space agency director, I want to assess the risk of collision between satellites and debris in Earth's orbit.
- As a space debris mitigation expert, I want to develop strategies to remove or mitigate space debris.
- As a commercial satellite provider, I want to analyze the market demand for satellite services in different orbits.
- As a satellite manufacturer, I want to analyze the failure rates and lifespans of different satellite models.
- As a policy maker, I want to develop regulations and guidelines for satellite operators and space debris mitigation.

(b) Example 2.

**Database: NBA Player Statistics 2023**

2023_nba_player_stats

| Player | POS | Team | Age | GP | W | L | Min | PTS | FGM | FGA | FG% | 3PM | 3PA | 3P% | FTM | … |
|---|---|---|---|---|---|---|---|---|---|---|---|---|---|---|---|---|
| Jayson Tatum | SF | BOS | 25.0 | 74.0 | 52.0 | 22.0 | 2732.2 | 2225.0 | 727.0 | 1559.0 | 46.6 | 240.0 | 686.0 | 35.0 | 531.0 | … |
| Joel Embiid | C | PHI | 29.0 | 66.0 | 43.0 | 23.0 | 2284.1 | 2183.0 | 728.0 | 1328.0 | 54.8 | 66.0 | 200.0 | 33.0 | 661.0 | … |
| Luka Doncic | PG | DAL | 24.0 | 66.0 | 33.0 | 33.0 | 2390.5 | 2138.0 | 719.0 | 1449.0 | 49.6 | 185.0 | 541.0 | 34.2 | 515.0 | … |
| Shai Gilgeous-Alexander | PG | OKC | 24.0 | 68.0 | 33.0 | 35.0 | 2416.0 | 2135.0 | 704.0 | 1381.0 | 51.0 | 58.0 | 168.0 | 34.5 | 669.0 | … |
| … | … | … | … | … | … | … | … | … | … | … | … | … | … | … | … | … |

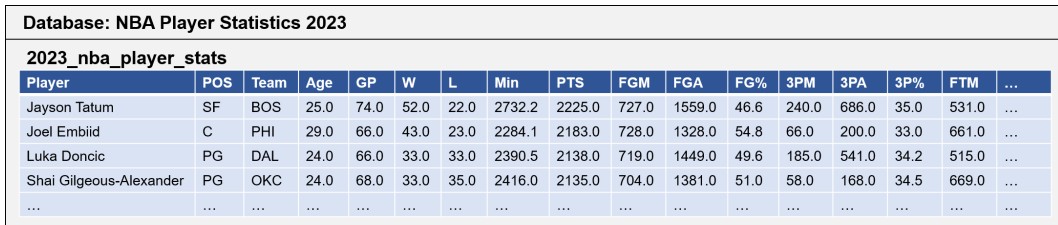

Questions:
- As a fantasy league player, I want to select players for my team based on their performance.
- As a school counselor or coach, I want to advise student-athletes on their career prospects in basketball.
- As a sports analyst, I want to use player statistics to identify trends and patterns in the sport of basketball.

(c) Example 3.

Figure 8: Examples of input databases and their associated queries.

**As a game developer, I want to balance the stats of different Pokemon to ensure fair gameplay.**

| SFT | RLHF |
|---|---|
| Findings
1. There are 8 attributes (height, weight, hp, attack, defense, sp atk, sp def, speed) in the dataset that have strong positive or negative correlations, indicating that certain attributes are related.
2. The pairs of attributes with strong positive or negative correlations are as follows:
    - height and weight (correlation: 0.65)
    - hp and attack (correlation: 0.45)
    - hp and defense (correlation: 0.44)
    - hp and speed (correlation: 0.34)
3. The pairs of attributes with strong negative correlation include:
    - sp atk and defense (correlation: -0.6)
    - sp def and speed (correlation: -0.5)

Suggestions
1. For balance purposes, you could consider assigning a higher weight to taller Pokemon and a higher percentage of the total weight for Pokemon, as height and weight are strongly correlated with each other.
2. To ensure that the gameplay remains diverse and fair, you should also assign appropriate attack, defense, sp atk, sp def, and speed values to the Pokemon, taking into consideration their correlations with the other attributes.
3. Since the Pokemon attributes are strongly correlated, consider implementing a more complex system that takes advantage of this relationship, such as a network or genetic algorithm-based approach to balance the attributes of diverse Pokemon species. | Findings
1. The majority of Pokemon are Water (38.13%), followed by Normal (24.72%) and Grass (13.82%).
2. Fire, Psychic, and Steel types are relatively less frequent in the dataset.
3. Types with high positive correlations with main stats (HP, attack, defense, and speed) include Water, Normal, and Grass types, while types with high negative correlations include Fire, Psychic, and Steel types.

Suggestions
1. Consider balancing the game around the most common types (Water, Normal, and Grass) to make the gameplay more diverse and interesting for different types of strategies.
2. Since types with high positive correlations have higher HPs, attacks, and speeds, it may be beneficial to focus on the Pokemon with those stats to balance the game accordingly.
3. Monitor the performance of high-positive correlation types (Water, Normal, and Grass) to ensure their representation in the game doesn't lead to an imbalance in terms of stats or gameplay strategy.
4. Monitor the performance of high-negative correlation types (Fire, Psychic, and Steel) to ensure their representation in the game isn't detrimental due to a potential imbalance in game mechanics. |

Figure 9: **Case study.**

I have a database of [database title]. I am a stakeholder and I am analyzing the database to make a decision. Who am I and what decision might it be? List 10 possibilities in a numbered list.

Each point should introduce who I am and briefly explain my intention in this format: As a/the [who I am], I want to [explain my intention]

Examples:

Based on the extracurricular activities database:
1. As the dean of student affairs, I want to decide on extracurricular activities to promote or cut
2. As the department head, I want to decide on faculty advisor assignments
3. As the school administrator, I want to review and revise faculty activity engagement

Based on a diabetes database:
1. As a healthcare policy maker, I want to decide on healthcare resource allocation
2. As a NIH official, I want to decide on medical research funding
3. As a health insurance actuary, I want to improve health insurance pricing strategy
4. As a health provider, I want to decide on patient care and treatment

Based on an allergy database:
1. As a catering manager, I want to plan meal options
2. As the school principal, I want to plan allergy awareness programs
3. As an administrator in the Student Affairs or Housing department, I want to decide on housing assignments
4. As the school administrator, I want to improve campus emergency preparedness
5. As the school principal, I want to develop policies for allergy accommodations

Based on a Home Equity Line of Credit (HELOC) product database, you can:
1. As the credit risk manager, I want to modify the credit underwriting policy

The database is as follows:

Database `[title]` has [x] tables. Table names are: [aaa], [bbb], [ccc]

Table `[caption]` has [x] rows and [y] columns. Column are:
`[column name]`, example values: [value 1], [value 2], [value 3], [value 4], [value 5]
...

Table 9: Prompt for **query collection**.

I will examine the relationships from the correlation matrix, including the most attended concerts and the conductor's experience, as well as the orchestras with the highest official ratings and their conductors.

```python
# Get the top 3 most attended concerts
top_attended = merged_df.nlargest(3, "Attendance")

# Get conductor names and age with most attended concerts
top_conductors = top_attended[["Name", "Age", "Year_of_Work", "Attendance"]].sort_values("Attendance",
ascending=False)

print("Top 3 Conductors with Most Attended Concerts:")
print(top_conductors)
print()

# Get top 3 orchestras with the highest official ratings
top_rated_orchestras = merged_df.nlargest(3, "Official_ratings_(millions)")

# Get conductor names and orchestras with the highest official ratings
top_rated_conductors = top_rated_orchestras[["Orchestra", "Name", "Age", "Year_of_Work",
"Official_ratings_(millions)"]].sort_values("Official_ratings_(millions)", ascending=False)

print("Top 3 Orchestras with Highest Official Ratings and Their Conductors:")
print(top_rated_conductors)
print()
```

Contribution RM: 5.05          Regularization RM: 5.15          Two RMs combined: 5.10

(a) A good case that receives high scores from both contribution RM and regularization RM.

Next, I will join the tables 'Order_Deliveries', 'Actual_Orders', 'Products', and 'Customers' to create a single DataFrame with the necessary information to analyze the most popular products across delivery routes.

```python
# Join tables
orders_deliveries = db["Order_Deliveries"]
actual_orders = db["Actual_Orders"]
products = db["Products"]
customers = db["Customers"]

# Merge tables
merged_df = pd.merge(orders_deliveries, actual_orders, on="actual_order_id")
merged_df = pd.merge(merged_df, products, left_on="actual_order_id", right_on="product_id")
merged_df = pd.merge(merged_df, customers, left_on="driver_employee_id", right_on="customer_id")

# Display a sample of 5 rows
print(merged_df.head())
```

Contribution RM: -2.63          Regularization RM: 6.59          Two RMs combined: 1.98

(b) A bad case that receives low score from contribution RM and high score from regularization RM.

Based on the analysis, provide a conclusion summarizing the findings and suggesting possible reasons for the high resettlement rates in the top 5 countries.

```python
print("Conclusion:")
print("The average resettlement rate in the top 5 countries is 593.8%, which is quite high compared to the
average resettlement rate of 553.8% in the entire dataset.")
print("This is due to the high resettlement rates in the top 5 countries, which may be due to the following
reasons:")
print("1. The top 5 countries have a higher population, which may indicate a higher demand for migration.")
print("2. The top 5 countries have specialized in Industry or Commerce, which might require a high number of
workers to be migrated.")
print("3. The top 5 countries have a significant history of migration, which may indicate a diverse population
and a need for migration.")
```

Contribution RM: 8.35          Regularization RM: -5.87          Two RMs combined: 1.24

(c) A reward hacking case that receives high score from contribution RM and low score from regularization RM.

Figure 10: **Qualitative examples** of code generations, and their scores assigned by reward models.

---

I have a database of [database title]. As a [stakeholder role], I want to [describe intention].

Given below two findings/conclusions, which one is more helpful to my analysis?
* [answer bullet point 1]
* [answer bullet point 2]

Your response should be in the following format:
* Reasoning: <explain your reasoning here>
* Answer: <repeat the more helpful finding here>

Table 10: Prompt for **helpfulness annotation collection**.

---

I have a database of [database title]. As a [stakeholder role], I want to [describe intention].

I have hired two data analysts to perform the analysis, and they gave me two different reports (listed below). Each report consists of two lists, one for findings and one for suggestions. Which one is more helpful to my analysis? When evaluating helpfulness, you should consider the following three rubrics in decreasing priority: (1) relevance to my analysis goal; (2) insightfulness; and (3) diversity of perspectives, especially for suggestions.

Your response should be in the following format. Note: <answer> should be either Report-1 or Report-2
* Answer: <answer>
* Reasoning: <explain your reasoning here>

The reports are as follows:

# Report-1

[report 1]

# Report-2

[report 2]

Table 11: Prompt for **helpfulness evaluation**.

