# OpenReview forum: "DACO: Towards Application-Driven and Comprehensive Data Analysis via Code Generation"
_NeurIPS.cc/2024/Datasets_and_Benchmarks_Track — NeurIPS 2024 Track Datasets and Benchmarks Poster_

### Official Review · Reviewer_dx7U · 2024-07-21
**Data analysis dataset, data leakage issues in annotation and creation**

**Rating:** 6
**Confidence:** 4
**Clarity:** Yes

**Review:**

This paper describes a DACO dataset, which consists of 440 databases of tables with 1942 corresponding queries that can help answer questions over the tables.  The focus in this work is notable for the fact that the queries were designed not to be simple fact finding over the data, but requires a chain of reasoning steps over the data (with code necessary to interact with the model).  This is indeed a very desirable property of the dataset.  A significant concern though is the reliance on the GPT family of models both to generate the dataset and to evaluate it.

**Strengths:**

1.  Useful collection of queries over datasets that require complex data analysis, using code and multiple reasoning steps.
2.  Some manual annotation of queries by humans about the steps in the queries, with human refinement over automated annotations.

**Additional Feedback:**

Please see my questions above.

**Correctness:**

Why not construct the benchmark the reverse way - which is look at code that does manipulate the datasets in Kaggle and have a code model describe the steps that the code is performing, plus what the overall query might be?

**Documentation:**

Yes

**Ethics:**

I am not clear about the licenses of the datasets from Kaggle.  Is it legal to create such a dataset?

**Limitations:**

No.

**Opportunities For Improvement:**

I'm concerned about using the same family of models (GPT various versions) for creating the dataset annotations, as well as judging the quality of the annotations.  This is clearly quite problematic.

Also, despite the filtering it seems odd that tables with 2 rows were included in the dataset - how is it possible to come up with a query over a table with 2 rows.

Finally, for datasets with up to 15 tables, what attempt was made to ensure that many of the tables were analyzed in the data analysis steps rather than a single one (given that data prep is considered to be the most tedious step in the data analysis process).

**Relation To Prior Work:**

Yes, as far as I know.

**Summary And Contributions:**

This paper describes a DACO dataset, which consists of 440 databases of tables with 1942 corresponding queries that can help answer questions over the tables.  The focus in this work is notable for the fact that the queries were designed not to be simple fact finding over the data, but requires a chain of reasoning steps over the data (with code necessary to interact with the model).  This is indeed a very desirable property of the dataset.  A significant concern though is the reliance on the GPT family of models both to generate the dataset and to evaluate it.

---

> ### Author Rebuttal · Authors · 2024-08-17
>
> Thank you for the insightful comments! We appreciate your recognition of our study's strengths, including useful data collection and manual refinement of the dataset. We would like to address the weaknesses as follows:
>
> > Concern about using the GPT family of models for both creating the dataset annotations and judging the quality of the annotations.
>
> We use GPT-4 to create queries and automatic annotations because of its strong coding and reasoning capabilities. **However, we do not use GPT models to evaluate or verify the quality of the annotations.** Instead, we take the following measures to ensure the quality of the generated dataset:
> 1. Filter the queries through human annotation, achieving a Cohen's kappa of 0.62,
> 2. Curate the condensed but high-quality test set through human annotation, achieving a Cohen’s kappa of 0.67.
> The high human annotation agreement verifies the quality of the resulting dataset.
>
> We adopt LLM for automatic evaluation due to the complexity of evaluating answer helpfulness. Prior work has shown that LLMs can reliably evaluate text generation quality based on task description and judging criteria [1].
>
> To mitigate the potential bias caused by using ChatGPT evaluators, which may favor the GPT family, we evaluate model helpfulness on $test^H$ with multiple LLM evaluators, including:
> | | Evaluator  | GPT-3.5 (as in the paper) | GPT4o-mini | Claude-3.5-Sonnet | Llama-8B |
> |-----------------|---------|------------|--------|----------|---|
> | **Model** | | | | |
> | Tapas     |      | 24.50   | 15.0       | 15.0   | 19.50    |
> | Tapex      |     | 6.00    | 9.50       | 3.50   | 14.00    |
> | ChatGPT w. code gen | | 27.27   | 18.69      | 19.19  | 26.26    |
> | GPT4 w. code gen  |  | 41.88   | 45.31      | 39.58  | 46.88    |
> | SFT w. code gen  |  | 8.79    | 5.76       | 7.05   | 16.67    |
> | FG-RLHF    |     | 21.05   | 13.82      | 7.24   | 16.45    |
>
> We observe that **different LLM evaluators show a moderate to substantial inter-annotator agreement of 0.45 Cohen’s kappa, and a very high Spearman correlation of 0.90 in model performance ranking**. These results suggest the robustness of LLM evaluation. To further alleviate the bias, we will update the helpfulness scores in the next revision into the average scores evaluated by LLMs from different families.
>
> > Question about tables with 2 rows
>
> The `# rows` and `# columns` values reported in Table 1 refer to the number of rows and columns in each individual table, not in each database. To clarify, each database can contain multiple tables. Sorry for the unclear presentation. Small tables with only 2 rows, like the one shown below, typically summarize key information of the database, and are always accompanied by larger tables with more detailed data.
> | product_id | name | price | billing_cycle |
> |---|---|---|---|
> | 0 | prd_1 | annual_subscription | 1200 | 12 |
> | 1 | prd_2 | monthly_subscription | 125 | 1 |
>
> Below is a clearer presentation of statistics. The `# rows` and `# columns` values of databases are calculated by summing across all tables within the database. **Notably, only 0.2% databases have less than 5 rows, and only 0.9% databases have less than 10 rows.**
> | | Med. | Max | Min |
> |---|---|---|---|
> |**Database:**| | | |
> | # tables | 1 | 15 | 1 |
> | # columns | 15 | 96 | 3 |
> | # rows | 572 | 67.2k | 4 |
> |**Table:**| | | |
> | # columns | 6 | 50 | 3 |
> | # rows | 20 | 19.2k | 2 |
>
> > How to ensure many tables were analyzed rather than a single one
>
> We take the following measures during query collection:
> 1. **Collecting comprehensive queries**: We collect queries that require cross-table analysis. We prompt GPT-4 to generate high-level queries driven by real-world applications, and use human annotation to filter out overly specific queries.
> 2. **Collecting diverse queries**: We generate multiple queries targeting different tables within the same database. We encourage output diversity when prompting GPT-4 and provide in-context examples with diverse outputs. Table 2 in the manuscript demonstrates the query diversity as 45.6% pairs of queries for the same database are highly different.
>
> To estimate the coverage of multiple tables, we examine how many tables are accessed by annotations of code annotation. **For databases with >5 tables, we find that each query covers 55.3% tables, and all queries for the same database cover 89.3%.** This confirms a broad coverage across tables. Moreover, some redundancy is both inevitable and beneficial, as it simulates the data redundancy in real-world settings.
>
> > Why not construct the benchmark from Kaggle code
>
> We agree that Kaggle code is a valuable resource. However, most of the codes focus on building and optimizing machine learning models rather than data analysis. Additionally, the codes are noisy, usually lacking structured formatting such as clear distinction between steps. While they could serve as a useful test set after manual cleaning, they are less suitable for constructing large-scale training dataset.
>
> > Licenses of Kaggle datasets
>
> Based on Kaggle’s policy, it is allowed to use and redistribute datasets as long as adhering to each dataset's specific license. All datasets utilized in this study are publicly accessible. A majority, 65%, are covered under various standardized licenses, all of which permit data redistribution for academic purposes. All these licenses allow data redistribution for academic purposes. The remaining 35% are licensed on an ad-hoc basis. We have manually reviewed their Kaggle descriptions to ensure there are no restrictions against using their data. However, since some datasets have limited description or licensing information, we will continue to monitor their status and will remove any dataset if issues arise. Detailed license information is in the attached pdf.
>
> ---
>
> References:
> 1. Liu, Yang, et al. "G-Eval: NLG Evaluation using Gpt-4 with Better Human Alignment." EMNLP 2023.

---

> > ### Comment · Reviewer_dx7U · 2024-08-19
> >
> > Thanks for the clarifications about the evaluations - that does indeed alleviate my concerns, and I have adjusted my score accordingly.

---

### Official Review · Reviewer_M7Gf · 2024-07-25

**Rating:** 6
**Confidence:** 4
**Correctness:** /
**Clarity:** The paper is well written.

**Review:**

This paper is significant in the field of data analysis. By employing innovative methods and high-quality datasets, it addresses the challenges that existing research faces in handling complex queries and demonstrates the potential of LLMs in data analysis. The proposed methods and experimental results hold substantial reference value for both academic research and practical applications.

**Strengths:**

- Introduction of DACO, a dataset created using a multi-turn prompting technique on a diverse set of real-world databases.
- Curation of a high-quality test set with comprehensive human annotations for refined benchmarking.
- Assessment of various models on the DACO dataset, including table QA models, prompt-based LLMs, and fine-tuned LLMs using SFT and RLHF.

**Additional Feedback:**

/

**Documentation:**

Yes.

**Ethics:**

No.

**Limitations:**

See "Opportunities For Improvement".

**Opportunities For Improvement:**

- Figure 3 seems a bit unclear.
- It is better to add an analysis of the labor costs.

**Relation To Prior Work:**

The paper clearly discusses how this work differs from previous contributions

**Summary And Contributions:**

This paper explore the task of data analysis, which involves systematic statistical and logical reasoning to derive insights from data.  The authors introduce a new dataset called DACO, constructed from diverse real-world databases with curated user queries.

---

> ### Author Rebuttal · Authors · 2024-08-17
>
> Thank you for the insightful comments! We appreciate your recognition of our study's strengths, including curating a high-quality test set and exploring various models. We would like to address the weaknesses as follows:
>
> > Clarification of Figure 3
>
> Figure 3 aims to illustrate the domain diversity of the databases by grouping them into 10 distinct topics using BERTopic. The upper half of Figure 3 displays the distribution of the ten topics, showing a long-tail effect. The lower half lists the keywords extracted by BERTopic for each of the top five topics to explain the content of each topic. We will improve the clarity of the figure captions in our next revision.
>
> > Request for an analysis of the labor costs
>
> The human labor involved in DACO dataset creation includes two steps:
> 1. **Query filtering**: As in L102-104, six annotators are trained to filter the automatically generated queries, and remove queries that are not driven by real-world applications or cannot be answered based on the database. A total of 4,593 queries were annotated, with each query reviewed by two annotators. Disagreements were resolved by post-annotation discussions. The removal agreement achieves a 0.62 Cohen’s kappa.
> 2. **Human refinement of test set**: As in L117-129, three expert annotators refined the automatically generated annotations to produce a high-quality test set. For each query, we generated three candidate annotations. The annotators densely annotated all three candidates and merged them into the final version. The three annotators worked on 59, 49, and 32 data points, respectively, from a pool of 100 unique data points. The overlapped ones are used for computing inter-annotator agreement, with a Cohen’s Kappa of 0.67.

---

### Official Review · Reviewer_wqLD · 2024-08-13
**The research makes a remarkable contribution to the field of automated data analysis through AI and code generation, though it could benefit from enhancements in the clarity of descriptions and methodology details.**

**Rating:** 6
**Confidence:** 4
**Clarity:** In general, the writing is satisfactory.

**Review:**

The quality of the work presented in the paper is satisfactory. The authors successfully communicate the objectives and implications of their research, making it accessible to readers with varying levels of expertise in the field. The originality is enough in its approach to automating data analysis using code generation. The significance lies in its potential to impact various applications that rely on database-based data analysis. By automating aspects of data interpretation and analysis, this research could enhance efficiency and accuracy in data-driven decision-making processes across different areas.

**Strengths:**

The paper proposes a novel use of code generation for automating data analysis, which is a significant step forward in applying AI in practical, real-world data scenarios.

The creation of the DACO dataset from a wide range of sources provides a robust platform for training and evaluating data analysis models.

The data and code associated with DACO are publicly available, encouraging further research and development in automated data analysis.

**Additional Feedback:**

Please see the Opportunities For Improvement and Limitations parts.

**Correctness:**

The claims made in the submission are correct and the dataset is constructed in a sound way. The benchmark and evaluation methods and experiment design are provided and performed correctly.

**Documentation:**

There is sufficient detail on data collection and organization, availability and maintenance, and ethical and responsible use.

**Ethics:**

No ethical concerns with the submission that warrant further discussion or review.

**Limitations:**

The evaluation metric, based on helpfulness, might be subjective and could vary significantly between different human annotators or even on different versions of ChatGPT. Some objective Question-Answer Pairs with exact answers such as multiple choice would be more subjective.

The performance of the models heavily relies on the quality of human annotations, which can be hard to scale in future works.

Limited comparison with other agent-based approaches. The paper could benefit from a more detailed comparison with other existing LLM agent models with code generation ability.

**Opportunities For Improvement:**

The source of 440 databases and the query generation can be improved with more clear descriptions. For example, to present a few examples of databases and queries in the appendix, which will greatly benefit the readers.

In fine-tuned LLMs of experiments, the CodeGeeX2-6B with RLHF is confusing without enough implementation details. I am wondering how the RLHF is performed in this setting, and would suggest the authors present the RLHF samples for training.

**Relation To Prior Work:**

It is clearly discussed how this work differs from previous contributions.

**Summary And Contributions:**

This paper presents a challenging and under-explored task in data analysis, which entails breaking down a user query into various perspectives, linking each perspective to the input data, and conducting logical and mathematical reasoning. This paper also introduces a new dataset called DACO (Data Analysis via Code Generation) that contains databases, query-answer pairs and a test set with human annotations to support this task. To prove the effectiveness of this dataset, this paper evaluates a diverse set of models on this dataset. These contributions make remarkable steps toward automating data analysis tasks, potentially transforming how insights are derived from large-scale data in various real-world applications.

---

> ### Author Rebuttal · Authors · 2024-08-17
>
> Thank you for the insightful comments! We appreciate your recognition of our study's strengths, mainly the important practical applications of the dataset. We would like to address the weaknesses as follows:
>
> > More examples of databases and queries
>
> Thank you for the suggestions. We have included a few examples in the pdf attached here, and we will include them into the Appendix in the next revision.
>
> > Clarification of RLHF
>
> The descriptions of RLHF are in L173-191. Sorry for any lack of clarity. Briefly, our RLHF model is trained to optimize scores from a reward model (RM), which is trained to assign higher scores to more helpful answers. The training data for the reward model is pairwise comparison of model generated final-answers ranked by ChatGPT.
>
> However, we observe that a single reward for the final answers was insufficient for training the intermediate coding steps, often resulting in overly short codes (e.g. only printing one or two statistics) producing insufficient information for answering the question. Following the fine-grained RLHF method [1], we handcraft two additional reward models: contribution RM and regularization RM. Contribution RM rewards codes that significantly contribute to the final answer, while regularization RM penalizes codes that exploit contribution RM. Training data for these two reward models are automatically synthesized, as detailed in L185-191.
>
> We will clarify these descriptions and add examples to demonstrate the behavior of each reward model in the next revision.
>
> > Evaluation metric based on helpfulness is subjective
>
> The subjective nature of this evaluation metric is inherent to the task. Given the complexity of data analysis questions, open-ended answers are crucial to present different perspectives, explain the reasoning process, and provide additional insights. While an alternative task such as question answering with fixed answers could simplify the evaluation, it could not well simulate this practical but challenging task.
>
> To mitigate subjectivity, we carefully define *helpfulness* as: (1) relevance to the query, (2) effective and insightful data interpretation, and (3) diversity in terms of analysis perspectives. **With a rigorous definition of helpfulness, both human evaluators and LLM evaluators have demonstrated a high level of agreement.**
>
> As in Table 4, the two human evaluation approaches - pairwise comparison and point-wise evaluation - both achieve high agreement of 0.62 and 0.65 Cohen’s kappa respectively.
>
> Additionally, we re-evaluate the model helpfulness on test^H with multiple LLM evaluators, as shown below:
> | | Evaluator  | GPT-3.5 (as in the paper) | GPT4o-mini | Claude-3.5-Sonnet | Llama-8B |
> |---|---|---|---|---|---|
> | **Model** | | | | |
> | Tapas     |      | 24.50   | 15.0       | 15.0   | 19.50    |
> | Tapex      |     | 6.00    | 9.50       | 3.50   | 14.00    |
> | ChatGPT w. code gen | | 27.27   | 18.69      | 19.19  | 26.26    |
> | GPT4 w. code gen  |  | 41.88   | 45.31      | 39.58  | 46.88    |
> | SFT w. code gen  |  | 8.79    | 5.76       | 7.05   | 16.67    |
> | FG-RLHF    |     | 21.05   | 13.82      | 7.24   | 16.45    |
>
> We observe that **different LLM evaluators show a moderate to substantial inter-annotator agreement of 0.45 Cohen’s kappa, and a very high Spearman correlation of 0.90 in model performance ranking**. This demonstrates the robustness of LLM evaluators, which is consistent of recent findings that LLM is able to reliably perform evaluation when provided with clear task descriptions and judging criteria definitions [2].
>
> > Model performance relies on human annotations, and thus is not scalable
>
> Our manual annotation efforts mainly focus on curating the high-quality test set. The collection of training set is largely automated, requiring limited human intervention to filter automatically generated queries, which better guarantees scalability.
>
> > Comparison with agent-based methods
>
> To clarify, our code generation framework involves iteratively planning, generating code, and then executing the code, which follows the ReAct method [3]. In this context, our framework can be considered as an adaptation of ReAct tailored to our task.
>
> Although there are some agent-based approaches that focus on code generation [4,5], they cannot be directly applied to our task. These methods typically focus on producing and refining a working code for a given problem, while our task requires dynamically exploring the database to gather information and perform reasoning accordingly before answering the question.
>
> However, it is feasible to adapt general-domain agents to our task, such as agents with tool crafting [6,7] and reflection [8,9]. The adaptation would require re-defining action space into code generation, and observation space into execution results. While we have not conducted these experiments in the current work, we will include discussions on this possibility in our next revision.
>
> ---
>
> References:
> 1. Wu, Zeqiu, et al. "Fine-grained human feedback gives better rewards for language model training." NeurIPS 2024.
> 2. Liu, Yang, et al. "G-Eval: NLG Evaluation using Gpt-4 with Better Human Alignment." EMNLP 2023.
> 3. Yao, Shunyu, et al. "ReAct: Synergizing Reasoning and Acting in Language Models." ICLR 2023.
> 4. Huang, Dong, et al. "Agentcoder: Multi-agent-based code generation with iterative testing and optimisation." arXiv 2312.13010.
> 5. Zhang, Kechi, et al. "Codeagent: Enhancing code generation with tool-integrated agent systems for real-world repo-level coding challenges." arXiv 2401.07339.
> 6. Cai, Tianle, et al. "Large Language Models as Tool Makers." ICLR 2024.
> 7. Yuan, Lifan, et al. "CRAFT: Customizing LLMs by Creating and Retrieving from Specialized Toolsets." ICLR 2024.
> 8. Shinn, Noah, et al. "Reflexion: Language agents with verbal reinforcement learning." NeurIPS 2024.
> 9. Madaan, Aman, et al. "Self-refine: Iterative refinement with self-feedback." NeurIPS 2024.

---

### Decision · Program_Chairs · 2024-09-26

**Decision:**

Accept (Poster)

**Comment:**

This study presents DACO, a dataset designed to address the challenge of real data-driven code generation. It introduces a well-curated collection spanning over 400 databases, with both manual and automatic curation processes. Reviewers consistently praised the dataset's strength and quality, despite minor limitations such as limited evaluations. I recommend accepting it as a poster.